# Meta-Reinforcement Learning of Structured Exploration Strategies

**Abhishek Gupta, Russell Mendonca, YuXuan Liu, Pieter Abbeel, Sergey Levine**
Department of Electrical Engineering and Computer Science
University of California, Berkeley
{abhigupta, pabbeel, svlevine}@eecs.berkeley.edu
{russellm, yuxuanliu}@berkeley.edu

## Abstract

Exploration is a fundamental challenge in reinforcement learning (RL). Many current exploration methods for deep RL use task-agnostic objectives, such as information gain or bonuses based on state visitation. However, many practical applications of RL involve learning more than a single task, and prior tasks can be used to inform how exploration should be performed in new tasks. In this work, we study how prior tasks can inform an agent about how to explore effectively in new situations. We introduce a novel gradient-based fast adaptation algorithm – model agnostic exploration with structured noise (MAESN) – to learn exploration strategies from prior experience. The prior experience is used both to initialize a policy and to acquire a latent exploration space that can inject structured stochasticity into a policy, producing exploration strategies that are informed by prior knowledge and are more effective than random action-space noise. We show that MAESN is more effective at learning exploration strategies when compared to prior meta-RL methods, RL without learned exploration strategies, and task-agnostic exploration methods. We evaluate our method on a variety of simulated tasks: locomotion with a wheeled robot, locomotion with a quadrupedal walker, and object manipulation.

## 1   Introduction

Deep reinforcement learning methods have been shown to learn complex tasks ranging from games [17] to robotic control [14, 20] with minimal supervision, by simply exploring the environment and experiencing rewards. As tasks become more complex or temporally extended, simple exploration strategies become less effective. Prior works have proposed guiding exploration based on criteria such as intrinsic motivation [23, 26, 25], state-visitation counts [16, 27, 2], Thompson sampling and bootstrapped models [4, 18], optimism in the face of uncertainty [3, 12], and parameter space exploration [19, 8]. These exploration strategies are largely task agnostic, in that they aim to provide good exploration without exploiting the particular structure of the task itself.

However, an intelligent agent interacting with the real world will likely need to learn many tasks, not just one, in which case prior tasks should be used to inform how exploration in new tasks should be performed. For example, a robot that is tasked with learning a new household chore likely has prior experience of learning other related chores. It can draw on these experiences to decide how to explore the environment to acquire the new skill more quickly. Similarly, a walking robot that has previously learned to navigate different buildings doesn't need to reacquire the skill of walking when it must learn to navigate through a maze, but simply needs to explore in the space of navigation strategies.

In this work, we study how experience from multiple distinct but related prior tasks can be used to autonomously acquire directed exploration strategies via meta-learning. Meta-learning, or learning to learn, refers to the problem of learning strategies which can adapt quickly to novel tasks by using

prior experience on different but related tasks [23, 28, 10, 29, 1, 21, 22]. In the context of RL, meta-learning algorithms typically fall into one of the following categories - RNN based learners [5, 30] and gradient descent based learners [6, 15].

RNN meta-learners address meta-RL by training recurrent models [5, 30] that ingest past states, actions, and rewards, and predict new actions that will maximize rewards, with memory across several episodes of interaction. These methods are not ideal for learning to explore. First, good exploration strategies are qualitatively different from optimal policies: while an optimal policy is typically deterministic in fully observed environments, exploration depends critically on stochasticity. Methods that simply recast the meta-RL problem into an RL problem generally acquire behaviors that exhibit insufficient variability to explore effectively in new settings for difficult tasks. The same policy has to represent highly exploratory behavior *and* adapt very quickly to optimal behavior, which becomes difficult with typical time-invariant representations for action distributions. Second, these methods aim to learn the entire "learning algorithm," using a recurrent model. While this allows them to adapt very quickly, via a single forward pass of the RNN, it limits their asymptotic performance when compared to learning from scratch, since the learned "algorithm" (i.e., RNN) generally does not correspond to a convergent iterative optimization procedure, and is not guaranteed to keep improving.

Gradient descent based meta-learners such as model-agnostic meta-learning (MAML) [6], directly train for model parameters that can adapt quickly with *gradient descent* for new tasks. These methods have the benefit of allowing for similar asymptotic performance as learning from scratch, since adaptation is performed using gradient descent, while also enabling acceleration from meta-training. However, our experiments show that MAML alone is not very effective at learning to explore, due to the lack of structured stochasticity in the exploration strategy.

We aim to address these challenges by devising a meta-RL algorithm that adapts to new tasks by following the policy gradient, while also injecting learned structured stochasticity into a latent space to enable effective exploration. Our algorithm, which we call model agnostic exploration with structured noise (MAESN), uses prior experience both to initialize a policy and to learn a latent exploration space from which it can sample temporally coherent structured behaviors. This produces exploration strategies that are stochastic, informed by prior knowledge, and more effective than random noise. Importantly, the policy and latent space are *explicitly* trained to adapt quickly to new tasks with the policy gradient. Since adaptation is performed by following the policy gradient, our method achieves at least the same asymptotic performance as learning from scratch (and often performs substantially better), while the structured stochasticity allows for randomized but task-aware exploration. Latent space models have been explored in prior works [9, 7, 13], though not in the context of meta-learning or learning exploration strategies. These methods do not explicitly train for fast adaptation, and comparisons in Section 4 illustrate the advantages of our method.

Our experimental evaluation shows that existing meta-RL methods, including MAML [6] and RNN-based algorithms [5, 30], are limited in their ability to acquire complex exploratory policies, likely due to limitations on their ability to acquire a strategy that is both stochastic and structured with policy parameterizations that can only introduce time-invariant stochasticity into the action space. While in principle certain RNN based architectures could capture time-correlated stochasticity, we find experimentally that current methods fall short. Effective exploration strategies must select randomly from among the *potentially useful* behaviors, while avoiding behaviors that are highly unlikely to succeed. MAESN leverages this insight to acquire significantly better exploration strategies by incorporating learned time-correlated noise through its meta-learned latent space, and training both the policy parameters and the latent exploration space explicitly for fast adaptation. In our experiments, we find that we are able to explore coherently and adapt quickly for a number of simulated manipulation and locomotion tasks with challenging exploration components. Some previous works

One natural question that arises with meta-learning exploration is: if our goal is to learn exploration strategies that solve challenging tasks with sparse or delayed rewards, how can we solve the diverse and challenging tasks at meta-training time to acquire those strategies in the first place? One approach that we can take with MAESN is to use dense or shaped reward tasks to meta-learn exploration strategies that work well for sparse or delayed reward tasks. In this setting, we assume that the meta-training tasks are provided with well-shaped rewards (e.g., distances to a goal), while the more challenging tasks that will be seen at meta-test time will have sparse rewards (e.g., an indicator for being within a small distance of the goal). As we will see in Section 4, this enables MAESN to solve

challenging tasks significantly better than prior methods at meta-test time for task families where existing meta-RL methods cannot meta-learn effectively from only sparse rewards.

## 2 Preliminaries: Meta-Reinforcement Learning

In meta-RL, we consider a distribution $\tau_i \sim p(\tau)$ over tasks, where each task $\tau_i$ is a different Markov decision process (MDP) $M_i = (S, A, P_i, R_i)$, with state space $S$, action space $A$, transition distribution $P_i$, and reward function $R_i$. The reward function and transitions vary across tasks. Meta-RL aims to to learn a policy that can adapt to maximize the expected reward for novel tasks from $p(\tau)$ as efficiently as possible.

We build on the gradient-based meta-learning framework of MAML [6], which trains a model in such a way that it can adapt quickly with standard gradient descent, which in RL corresponds to the policy gradient. The meta-training objective for MAML can be written as

$$\max_{\theta} \sum_{\tau_i} \mathbb{E}_{\pi_{\theta_i'}} \left[ \sum_t R_i(s_t) \right] \qquad \theta_i' = \theta + \alpha \mathbb{E}_{\pi_\theta} \left[ \sum_t R_i(s_t) \nabla_\theta \log \pi_\theta(a_t|s_t) \right] \qquad (1)$$

The intuition behind this optimization objective is that, since the policy will be adapted at meta-test time using the policy gradient, we can optimize the policy parameters so that one step of policy gradient improves its performance on any meta-training task as much as possible.

Since MAML reverts to conventional policy gradient when faced with out-of-distribution tasks, it provides a natural starting point for us to consider the design of a meta-exploration algorithm: by starting with a method that is essentially on par with task-agnostic RL methods that learn from scratch in the *worst* case, we can improve on it to incorporate the ability to acquire stochastic exploration strategies from experience, while preserving asymptotic performance.

## 3 Model Agnostic Exploration with Structured Noise

While meta-learning has been shown to be effective for fast adaptation on several RL problems [6, 5], the prior methods generally focus on tasks where exploration is trivial and a few random trials are sufficient to identify the goals of the task [6], or the policy should acquire a consistent "search" strategy, for example to find the exit in new mazes [5]. Both of these adaptation regimes differ substantially from stochastic exploration. Tasks where discovering the goal requires exploration that is both stochastic *and* structured cannot be easily captured by such methods, as demonstrated in our experiments. Specifically, there are two major shortcomings with these methods: (1) The stochasticity of the policy is limited to time-invariant noise from action distributions, which fundamentally limits the exploratory behavior it can represent. (2) For RNN based methods, the policy is limited in its ability to adapt to new environments, since adaptation is performed with a forward pass of the recurrent network. If this single forward pass does not produce good behavior, there is no further mechanism for improvement. Methods that adapt by gradient descent, such as MAML, simply revert to standard policy gradient and can make slow but steady improvement in the worst case, but do not address (1). In this section, we introduce a novel method for learning structured exploration behavior based on gradient based meta-learning which is able to learn good exploratory behavior *and* adapt quickly to new tasks that require significant exploration, without suffering in asymptotic performance.

### 3.1 Overview

Our algorithm, which we call model agnostic exploration with structured noise (MAESN), combines structured stochasticity with MAML. MAESN is a gradient-based meta-learning algorithm that introduces stochasticity not just by perturbing the actions, but also through a learned latent space which allows exploration to be time-correlated. Both the policy and the latent space are trained with meta-learning to explicitly provide for fast adaptation to new tasks. When solving new tasks at meta-test time, a different sample is generated from this latent space for each episode (and kept fixed throughout the episode), providing structured and temporally correlated stochasticity. Because of meta-training, the distribution over latent variables is adapted to the task *quickly* via policy gradient updates. We first show how structured stochasticity can be introduced through latent spaces, and then describe how both the policy and the latent space can be meta-trained to form our overall algorithm.

## 3.2 Policies with Latent State

Typical stochastic policies parameterize action distributions $\pi_\theta(a|s)$ in a way that is independent for each time step. This representation has no notion of temporally coherent randomness throughout the trajectory, since stochasticity is added independently at each step. Under this representation, additive noise is sampled independently for every time step. This limits the range of possible exploration strategies, since the policy essentially "changes its mind" about what it wants to explore at each time step. The distribution $\pi_\theta(a|s)$ is also typically represented with simple parametric families, such as unimodal Gaussians, which restrict its expressivity.

To incorporate temporally coherent exploration and allow the policy to model more complex time-correlated stochastic processes, we can condition the policy on per-episode random variables drawn from a learned latent distribution, as shown on the right. Since these latent variables are sampled only once per episode, they provide temporally coherent stochasticity. Intuitively, the policy decides only once what it will try to do in each episode, and commits to this plan. Since the random sample is provided as an input, a nonlinear neural network policy can transform this sample into arbitrarily complex distributions.

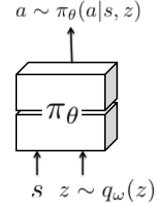

The resulting policies can be written as $\pi_\theta(a|s,z)$, where $z \sim q_\omega(z)$, and $q_\omega(z)$ is the latent variable distribution with parameters $\omega$. For example, in our experiments we consider diagonal Gaussian distributions of the form $q_\omega(z) = \mathcal{N}(\mu, \sigma)$, such that $\omega = \{\mu, \sigma\}$. Structured stochasticity of this form can provide more coherent exploration, by sampling entire behaviors or goals, rather than simply relying on independent random actions.

We discuss how to meta-learn latent representations and adapt quickly to new tasks. Related representations have been explored in prior work [9, 7] but simply inputting random variables into a policy does not by itself provide for rapid adaptation to new tasks. To achieve fast adaptation, we can incorporate meta-learning as discussed below.

## 3.3 Meta-Learning Latent Variable Policies

Given a latent variable conditioned policy as described above, our goal is to train it so as to capture coherent exploration strategies from a family of training tasks that enable *fast adaptation* to new tasks from a similar distribution. We use a combination of variational inference and gradient-based meta-learning to achieve this. Specifically, our aim is to meta-train the policy parameters $\theta$ so that they can make use of the latent variables to perform coherent exploration on a new task and the behavior can be adapted as fast as possible. To that end, we jointly learn a set of policy parameters and a set of latent space distribution parameters, such that they achieve

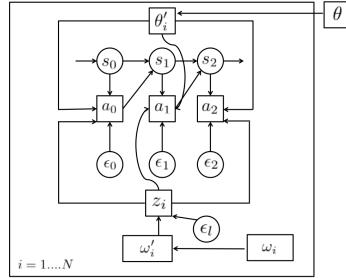

Figure 1: Computation graph for MAESN. Meta-learn pre-update latent parameters $\omega_i$, and policy parameters $\theta$, such that after a gradient step, the post-update latent parameters $\omega_i'$, policy parameters $\theta'$, are optimal for the task. The sampling procedure introduces time correlated noise.

optimal performance for each task *after* a policy gradient adaptation step. This procedure encourages the policy to actually make use of the latent variables for exploration. From one perspective, MAESN can be understood as augmenting MAML with a latent space to inject structured noise. From a different perspective it amounts to learning a structured latent space, similar to [9], but trained for quick adaptation to new tasks. While [6] enables quick adaptation for simple tasks, and [9] learns structured latent spaces, MAESN can achieve both structured exploration and fast adaptation. As shown in our experiments, neither of the prior methods alone effectively learn complex and stochastic exploration strategies.

To formalize the objective for meta-training, we introduce a model parameterization with policy parameters $\theta$ shared across all tasks, and per-task variational parameters $\omega_i$ for tasks $i = 1, 2..., N$, which parameterize a per-task latent distribution $q_{\omega_i}(z_i)$. We refer to $\theta, \omega_i$ as the pre-update parameters. Meta-training involves optimizing the pre-update parameters on a set of training tasks, so as to maximize expected reward *after* a policy gradient update. As is standard in variational inference, we also add to the objective the KL-divergence between the per-task pre-update distributions $q_{\omega_i}(z_i)$ and a prior $p(z)$, which in our experiments is simply a unit Gaussian. Without this additional loss,

the per-task parameters $\omega_i$ can simply memorize task-specific information. The KL loss ensures that sampling $z \sim p(z)$ for a new task at meta-test time still produces effective structured exploration.

For every iteration of meta-training, we sample from the latent variable conditioned policies represented by the pre-update parameters $\theta, \omega_i$, perform an "inner" gradient update on the variational parameters for each task (and, optionally, the policy parameters) to get the task-specific post-update parameters $\theta'_i, \omega'_i$, and then propagate gradients through this update to obtain a meta-gradient for $\theta$, $\omega_0, \omega_1, ..., \omega_N$ such that the sum of expected task rewards over all tasks using the post-update latent-conditioned policies $\theta'_i, \omega'_i$ is maximized, while the KL divergence of pre-update distributions $q_{\omega_i}(z_i)$ against the prior $p(z_i)$ is minimized. Note that the KL-divergence loss is applied to the pre-update distributions $q_{\omega_i}$, not the post-update distributions, so the policy can exhibit very different behaviors on each task after the inner update. Computing the gradient of the reward under the post-update parameters requires differentiating through the inner policy gradient term, as in MAML [6].

A concise description of the meta-training procedure is provided in Algorithm 1, and the computation graph representing MAESN is shown in Fig 1. The full meta-training problem can be stated mathematically as

$$\max_{\theta, \omega_i} \sum_{i \in tasks} E_{\substack{a_t \sim \pi(a_t|s_t; \theta'_i, z'_i) \\ z'_i \sim q_{\omega'_i}(.)}} \left[ \sum_t R_i(s_t) \right] - \sum_{i \in tasks} D_{KL}(q_{\omega_i}(.) \| p(z)) \tag{2}$$

$$\omega'_i = \omega_i + \alpha_\omega \circ \nabla_{\omega_i} E_{\substack{a_t \sim \pi(a_t|s_t; \theta, z_i) \\ z_i \sim q_{\omega_i}(.)}} \left[ \sum_t R_i(s_t) \right] \tag{3}$$

$$\theta'_i = \theta + \alpha_\theta \circ \nabla_\theta E_{\substack{a_t \sim \pi(a_t|s_t; \theta, z_i) \\ z_i \sim q_{\omega_i}(.)}} \left[ \sum_t R_i(s_t) \right] \tag{4}$$

The two objective terms are the expected reward under the *post update* parameters for each task and the KL-divergence between each task's pre-update latent distribution and the prior. The $\alpha$ values are per-parameter step sizes, and $\circ$ is an elementwise product. The last update (to $\theta$) is optional. We found that we could in fact obtain better results simply by omitting this update, which corresponds to meta-training the initial policy parameters $\theta$ simply to use the latent space efficiently, without training the parameters themselves explicitly for fast adaptation. Including the $\theta$ update makes the resulting optimization problem more challenging.

MAESN enables structured exploration by using the latent variables $z$, while explicitly training for fast adaptation via policy gradient. We could in principle train such a model without meta-training for adaptation at all, which resembles the model proposed by [9]. However, as we will show in our experimental evaluation, meta-training produces substantially better results.

Interestingly, during the course of meta-training, we find that the pre-update variational parameters $\omega_i$ for each task are usually close to the prior at convergence. This has a simple explanation: meta-training optimizes for *post-update* rewards, after $\omega_i$ have been updated to $\omega'_i$, so even if $\omega_i$ matches the prior, it does not match the prior after the inner update. This allows the learned policy to succeed on new tasks at meta-test time for which we do not have a good initialization for $\omega$, and have no choice but to begin with the prior, as discussed in the next section.

---

**Algorithm 1** MAESN meta-RL algorithm
---
1: Initialize variational parameters $\omega_i$ for each training task $\tau_i$
2: **for** iteration $k \in \{1, \dots, K\}$ **do**
3:      Sample a batch of $N$ training tasks from $p(\tau)$
4:      **for** task $\tau_i \in \{1, \dots, N\}$ **do**
5:          Gather data using the latent conditioned policy $\theta, (\omega_i)$
6:          Compute inner policy gradient on variational parameters via Equation (4) (optionally (5))
7:      **end for**
8:      Compute meta update on both latents and policy parameters by optimizing (3) with TRPO
9: **end for**
---

### 3.4 Using the Latent Space for Exploration

Let us consider a new task $\tau_i$ with reward $R_i$, and a learned model with policy parameters $\theta$. The variational parameters $\omega_i$ are specific to the tasks used during meta-training, and will not be useful for a new task. However, since the KL-divergence loss (Eqn 3) encourages the pre-update parameters to be close to the prior, all of the variational parameters $\omega_i$ are driven to the prior at convergence (Fig 5a). Hence, for exploration in a new task, we can initialize the latent distribution to the prior $q_\omega(z) = p(z)$. In our experiments, we use the prior with $\mu = 0$ and $\sigma = I$. Adaptation to a new task is then done by simply using the policy gradient to adapt $\omega$ via backpropagation on the RL objective, $\max_\omega E_{a_t \sim \pi(a_t|s_t,\theta,z), z \sim q_\omega(.)} \left[ \sum_t R(s_t) \right]$ where $R$ represents the sum of rewards along the trajectory. Since we meta-trained to adapt $\omega$ in the inner loop, we adapt these parameters at meta-test time as well. To compute the gradients with respect to $\omega$, we need to backpropagate through the sampling operation $z \sim q_\omega(z)$, using either likelihood ratio or the reparameterization trick(if possible). The likelihood ratio update is

$$\nabla_\omega \eta = E_{\substack{a_t \sim \pi(a_t|s_t;\theta,z) \\ z \sim q_\omega(.)}} \left[ \nabla_\omega \log q_\omega(z) \sum_t R(s_t) \right] \tag{5}$$

This adaptation scheme has the advantage of quick learning on new tasks because of meta-training, while maintaining good asymptotic performance since we are simply using the policy gradient.

## 4 Experiments

Our experiments aim to comparatively evaluate our meta-learning method and study the following questions: (1) Can meta-learned exploration strategies with MAESN explore coherently and adapt quickly to new tasks, providing a significant advantage over learning from scratch? (2) How does meta-learning with MAESN compare with prior meta-learning methods such as MAML [6] and RL$^2$ [5], as well as latent space learning methods [9]? (3) Can we visualize the exploration behavior and see coherent exploration strategies with MAESN? (4) Can we better understand which components of MAESN are the most critical? Videos and experimental details for all our experiments can be found at `https://sites.google.com/view/meta-explore/`

### 4.1 Experimental Details

During meta-training, the "inner" update corresponds to standard REINFORCE, while the meta-optimizer is trust region policy optimization(TRPO) [24]. Hyperparameters of each algorithm are mentioned in the supplementary materials, which were selected via a hyperparameter sweep (also detailed in the appendix). All experiments were initially run on a local 2 GPU machine, and run at scale using Amazon Web Services. While our goal is to adapt quickly with sparse and delayed rewards at meta-test time, this poses a major challenge at meta-training time: if the tasks themselves are too difficult to learn from scratch, they will also be difficult to solve at meta-training time, making it hard for the meta-learner to make progress. In fact, none of the methods we evaluated, including MAESN, were able to make *any* learning progress on the sparse reward tasks at meta-training time (refer to meta-training progress in supplementary materials Fig 2).

While this issue could potentially be addressed by using many more samples or existing task-agnostic exploration strategies during meta-training only, our method allows for a simpler solution. As discussed in Section 1, we can make use of shaped rewards during meta-training (both for our method and for baselines), while only the sparse rewards are used to adapt at meta-test time. As shown below, exploration strategies with MAESN meta-trained with reward shaping generalize effectively to sparse and delayed rewards, despite the mismatch in the reward function.

### 4.2 Task Setup

We evaluated our method on three task distributions $p(\tau)$. For each family of tasks we used 100 distinct meta-training tasks, each with a different reward function $R_i$. After meta-training on a particular distribution of tasks, MAESN is able to explore well and adapt quickly to tasks drawn from this distribution (with sparse rewards). The input state of the environments does not contain the goal – instead, the agent must explore different locations to locate the goal through exploration. The details of the meta-train and test reward functions can be found in the supplementary materials.

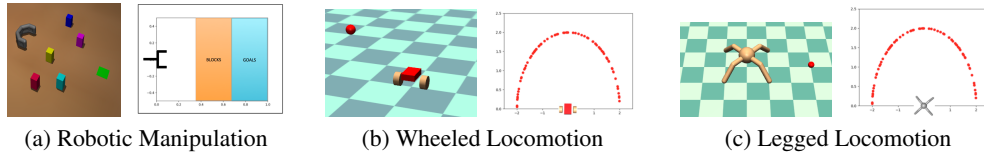

(a) Robotic Manipulation       (b) Wheeled Locomotion       (c) Legged Locomotion

Figure 2: Task distributions for MAESN. For each subplot, left shows the general task setup and right shows the distribution of tasks. For robotic manipulation, orange indicates block location region across tasks, and blue indicates the goal regions. For both locomotion tasks, the red circles indicate goal positions across tasks from the distribution.

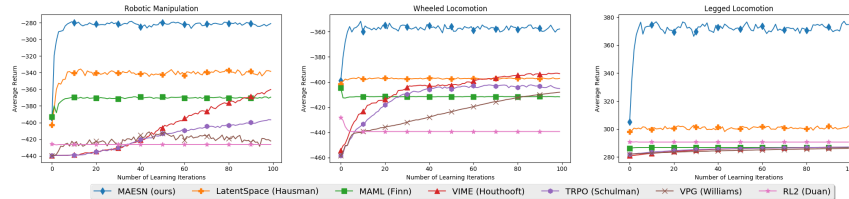

Figure 3: Learning progress on novel tasks with sparse rewards for wheeled locomotion, legged locomotion, and object manipulation. Rewards are averaged over 100 validation tasks, which have sparse rewards as described in supplementary material. MAESN learns significantly better policies, and learns much quicker than prior meta-learning approaches and learning from scratch.

**Robotic Manipulation.** The goal in these tasks is to push blocks to target locations with a robotic hand. Only one block (unknown to the agent) is relevant for each task, and that block must be moved to a goal location (see Fig. 2a). The position of the blocks and the goals are randomized across tasks. A coherent exploration strategy should pick random blocks to move to the goal location, trying different blocks on each episode to discover the right one. This task is generally representative of exploration challenges in robotic manipulation: while a robot might perform a variety of different manipulation skills, only motions that actually interact with objects in the world are useful for coherent exploration.

**Wheeled Locomotion.** We consider a wheeled robot which controls its two wheels independently to move to different goal locations. The task family is illustrated in Fig. 2b. Coherent exploration on this family of tasks requires driving to random locations in the world, which requires a coordinated pattern of actions that is difficult to achieve purely with action-space noise.

**Legged Locomotion.** To understand if we can scale to more complex locomotion tasks, we consider a quadruped ("ant") tasked to walk to randomly placed goals (see Fig. 2c). This task presents a further exploration challenge, since only carefully coordinated leg motion produces movement to different positions, so an ideal exploration strategy would always walk, but to different places.

### 4.3 Comparisons

We compare MAESN with $RL^2$ [5], MAML [6], simply learning latent spaces without fast adaptation(LatentSpace), analogously to [9]. For training from scratch, we compare with TRPO [24], REINFORCE [31], and training from scratch with VIME [11], a general-purpose exploration algorithm. Further details can be found in the supplementary materials.

In Figure 3, we report results for our method and prior approaches when adapting to new tasks at meta-test time, using sparse rewards. We plot the performance of all methods in terms of the reward (averaged across 30 validation tasks) that the methods obtain while adapting to tasks drawn from a test set of tasks. Our results on the tasks we discussed above show that MAESN is able to explore and adapt quickly on sparse reward environments. In comparison, MAML and $RL^2$ don't learn behaviors that explore as effectively. The pure latent spaces model (LatentSpace in Figure 3) achieves reasonable performance, but is limited in terms of its capacity to improve beyond the initial identification of latent space parameters and is not optimized for fast adaptation in the latent space. Since MAESN can train the latent space explicitly for fast adaptation, it achieves better results faster.

We also observe that, for many tasks, learning from scratch actually provides a competitive baseline to prior meta-learning methods in terms of asymptotic performance. This indicates that the task distributions are quite challenging, and simply memorizing the meta-training tasks is insufficient to succeed. However, in all cases, we see that MAESN is able to outperform learning from scratch and task-agnostic exploration in terms of both learning speed and asymptotic performance.

On the challenging legged locomotion task, which requires coherent walking behaviors to random locations in the world to discover the sparse rewards, we find that only MAESN is able to adapt effectively.

## 4.4  Exploration Strategies

To understand the exploration strategies learned by MAESN, we visualize the trajectories obtained by sampling from the meta-learned latent-conditioned policy $\pi_\theta$ with the latent distribution $q_\omega(z)$ set to the prior $\mathcal{N}(0, I)$. The resulting trajectories show the 2D position of the hand for the block pushing task and the 2D position of the center of mass for the locomotion tasks. Task distributions for each family of tasks are shown in Fig 2a, 2b, 2c. We can see from these trajectories (Fig 4) that learned exploration strategies explore in the space of coherent behaviors broadly and effectively, especially in comparison with random exploration and standard MAML.

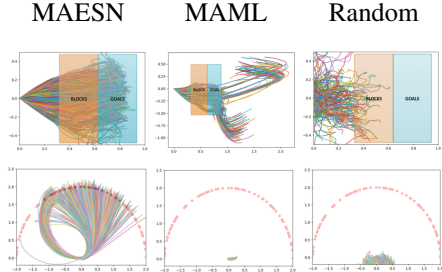

MAESN       MAML       Random

Figure 4: Plot of exploration behavior visualizing 2D position of the manipulator (for blockpushing) and CoM for locomotion for MAESN, MAML and random initialization. **Top:** Block Manipulation **Bottom:** Wheeled Locomotion. Goals indicated by the translucent overlays. MAESN captures the task distribution better than other methods.

## 4.5  Analysis of Structured Latent Space

We investigate the structure of the learned latent space in the manipulation task by visualizing pre-update $\omega_i = (\mu_i, \sigma_i)$ and post-update $\omega_i' = (\mu_i', \sigma_i')$ parameters for a 2D latent space. The variational distributions are plotted as ellipses. As can be seen from Fig 5a, pre-update parameters are all driven to the prior $\mathcal{N}(0, I)$, while the post-update parameters move to different locations in the latent space to adapt to their respective tasks. This indicates that the meta-training process effectively utilizes the latent variables, but also minimizes the KL-divergence against the prior, ensuring that initializing $\omega$ to the prior for a new task will produce effective exploration.

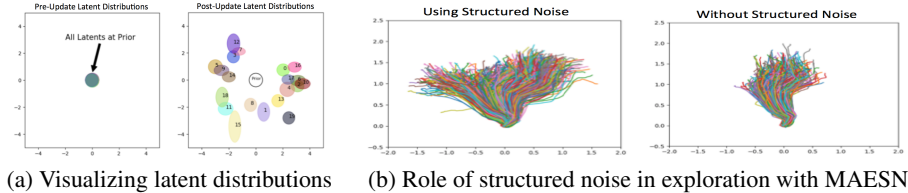

(a) Visualizing latent distributions        (b) Role of structured noise in exploration with MAESN

Figure 5: Analysis of learned latent space **(a)** Latent distributions in MAESN visualized for a 2D latent space.**Left:** Pre-update latents, **Right:** Post update latents. (Each number in the post-update plot corresponds to a different task.) **(b)** Visualization of exploration for legged locomotion **Left:** CoM visitations using structured noise. **Right:** CoM visitations with no structured noise. Increased spread of exploration and wider trajectory distribution suggests that structured noise *is* being used.

We also evaluate whether the noise injected from the latent space learned by MAESN is actually used for exploration. We observe the exploratory behavior displayed by a policy trained with MAESN when the latent variable $z$ is kept fixed, as compared to when it is sampled from the learned latent distribution. We can see from Fig. 5b that, although there is some random exploration even without latent space sampling, the range of trajectories is much broader when $z$ is sampled from the prior.

## 5  Conclusion

We presented MAESN, a meta-RL algorithm that explicitly learns to explore by combining gradient-based meta-learning with a learned latent exploration space. MAESN learns a latent space that can be used to inject temporally correlated, coherent stochasticity into the policy to explore effectively at

meta-test time. A good exploration strategy must randomly sample from among the *useful* behaviors, while omitting behaviors that are never useful. Our experimental evaluation illustrates that MAESN does precisely this, outperforming both prior meta-learning methods and learning from scratch, including methods that use task-agnostic exploration strategies. It's worth noting, however, that our approach is not mutually exclusive with these methods, and in fact a promising direction for future work would be to combine our approach with these methods [11].

## 6 Acknowledgements

The authors would like to thank Chelsea Finn, Gregory Kahn, Ignasi Clavera for thoughtful discussions and Justin Fu, Marvin Zhang for comments on an early version of the paper. This work was supported by a National Science Foundation Graduate Research Fellowship for Abhishek Gupta, ONR PECASE award for Pieter Abbeel, and the National Science Foundation through IIS-1651843 and IIS-1614653, as well as an ONR Young Investigator Program award for Sergey Levine.

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
