[Supplementary Material]

# Supplemental Materials

**Abhishek Gupta, Russell Mendonca, YuXuan Liu, Pieter Abbeel, Sergey Levine**
Department of Electrical Engineering and Computer Science
University of California, Berkeley
{abhigupta, pabbeel, svlevine}@eecs.berkeley.edu
{russellm, yuxuanliu}@berkeley.edu

## 1  Experimental Details

We built all of our implementation on the open source implementation of rllab Duan et al. (2016) and MAML Finn et al. (2017a). Our policies were all feedforward policies of 2 layers, with a hundred units each and ReLU nonlinearities. We performed meta-training for a single step of adaptation, though longer could be done in principle.

We found that to get MAESN to work well, meta-learning a per-parameter stepsize is crucial, rather than keeping step-size fixed. This has been found to help in prior work Li et al. (2017) as well.

## 2  Reward Functions

While training all tasks we used dense reward functions to enable meta-training as described in Section [4] of the paper. For each of the tasks, the dense rewards are given by

$$R_{block} = -\|x_{obj} - x_{goal}\|_2 \tag{1}$$

$$R_{wheeled} = -\|x_{com} - x_{goal}\|_2 \tag{2}$$

$$R_{ant} = -\|x_{com} - x_{goal}\|_2 \tag{3}$$

The test-time reward is sparser, provided only in a region around the target position. The sparse rewards for these tasks are given by

$$R_{block} = \begin{cases} -c_{max} & \|x_{obj} - x_{goal}\|_2 > 0.2 \\ -\|x_{obj} - x_{goal}\|_2 & \|x_{obj} - x_{goal}\|_2 \le 0.2 \end{cases}$$

$$R_{wheeled} = \begin{cases} -c_{max} & \|x_{com} - x_{goal}\|_2 > 0.8 \\ -\|x_{obj} - x_{goal}\|_2 & \|x_{obj} - x_{goal}\|_2 \le 0.8 \end{cases}$$

$$R_{ant} = \begin{cases} 4 - c_{max} & \|x_{obj} - x_{goal}\|_2 > 0.8 \\ 4 - \|x_{obj} - x_{goal}\|_2 & \|x_{obj} - x_{goal}\|_2 \le 0.8 \end{cases}$$

where $-c_{max}$ is an uninformative large negative constant reward. The reward is uninformative until the agent/object reach a threshold distance around the goal, and then the negative distance to the goal is subsequently provided as the reward function.

We built all of our implementation on the open source implementation of rllab Duan et al. (2016) and MAML Finn et al. (2017a). Our policies were all feedforward policies of 2 layers, with a hundred units each and ReLU nonlinearities. We performed meta-training for a single step of adaptation, though longer could be done in principle.

We found that to get MAESN to work well, meta-learning a per-parameter stepsize is crucial, rather than keeping step-size fixed. This has been found to help in prior work Li et al. (2017) as well.

# 3 Hyperparameters

We describe the hyper-parameters swept over for gradient based meta-learning methods. For meta-training we swept over:

- number of trajectories per task in [20 , 50] for MAESN, latentspace baseline and MAML

- latent parameter size in [2,4,8] for MAESN, latentspace baseline

- bias transform size in [2,4,8] for MAML

- kl weighting in [0.1, 0.5] for MAESN and in [0.001 , 0.01, 0.05] for latentspace baseline

- inner learning rate in [0.5 , 1] for MAESN and MAML

For each method, we selected hyperparameters which gave best train time performance and then ran meta-testing. Both meta-training and meta-testing were run with multiple seeds.

# 4 Ablation Study

Since MAESN introduces a number of components such as adaptive step size, a latent space for exploration to the framework of MAML, we perform ablations to see which of these make a major difference. Adding in a learned latent space (called bias transformation) has been explored before in Finn et al. (2017b) but the latent space was not stochastic, making it non-helpful for exploration.

Figure 1: Ablation study comparing adaptation performance on novel tasks of MAESN against a number of variants of MAML - using bias transformation, adaptive stepsize or a combination of both

We found that although adding in a bias transformation to MAML was helpful, it did not match the performance of MAESN. Variants considered are (1) standard MAML (2) MAML + bias transform + adaptive stepsize, adapting all parameters in the inner update (maml+Bias +allParameterAdaptation) (3) MAML + bias transform + adaptive stepsize, adapting only the bias parameters in the inner update (maml+Bias +onlyBiasAdaptation).

# 5 Meta-Training Performance

On meta-training with sparse reward, none of the methods learned to do anything as can be seen by curves of post-update reward on the wheeled locomotion environment. On meta-training with dense reward, both MAML and MAESN achieve quite good post-update reward as seen by the following curves. However, as seen from learned exploration schemes in the following section, we see that MAESN learns to explore while MAML does not, which enables MAESN to transfer better to new sparse reward tasks.

(a) Dense Reward

(b) Sparse Reward

Figure 2: Comparison between training with dense and sparse reward for the wheeled locomotion environment

# 6 MAESN Test-Time Trajectories

At test-time, MAESN not only gets high reward on the sparse reward tasks, but also completely solves them. Below are the trajectories when the agent is tested on a few goals representative of the overall task distribution.

(a) Left Goal

(b) Center Goal

(c) Right Goal

Figure 3: Sample ant test-time trajectories. The ant starts at the yellow circle, and the target is the green circle. (The solid plot is of the final position of the ant)