[Reviews · NeurIPS 2018]

Reviewer 1



Response to the rebuttal: Thanks for the useful clarifications and new experiments. I have increased my score rom a 4 to a 7. The main concern in my original review was that the reward signal was switched from dense to sparse rewards for evaluation but I am now convinced that it's a reasonable domain for analysis. I think an explicit discussion on switching the reward signal would be useful to include in the final version of the paper. --- This paper proposes a way of using MAML with policies in RL that have a stochastic latent state. They show how to use MAML to update the distribution of the latent state in addition to using standard MAML to update the parameters of the policy. They empirically show that these policies learn a reasonable exploration policy in sparse-reward manipulation and locomation domains. I like the semantics of using MAML to update the distribution of a policy's stochastic latent state. I see this as an implicit and tractable way of representing distributions over states that are worth exploring without needing to explicitly represent these distributions, which can quickly become intractable. I am not sure how to interpret the empirical results because if I understand correctly, all of the methods are trained in the dense reward setting (Fig 2 of the appendix) and then all of the results and analysis in the main paper use the sparse reward setting for validation, which are out-of-distribution tasks. When training in the dense reward setting, MAML outperforms the proposed method (MAESN) but in the sparse reward setting for validation, MAESN outperforms MAML. The paper claims that succeeding in the out-of-distribution sparse reward setting shows that MAESN is better than MAML at exploring and that MAML does not learn how to explore. It is not clear that out-of-distribution performance is indicative of explorative behavior or if it's just showing that a different policy is learned. I think this is especially important to consider and further analyze because MAML slightly outperforms MAESN for the training tasks with dense reward. I would be more convinced if the paper showed an experiment where MAESN outperforms MAML for training in a more complex task that requires exploration. Also, the reason training was done in the dense reward setting is because none of the methods are able to be trained from scratch in the sparse reward setting as shown in Figure 2b. Have you considered using the trained models from the dense reward domain to start training in the sparse reward domain?

Reviewer 2



This paper extends the MAML method by introducing structured randomness to the latent exploration space to perform more effective exploration. The experimental results show that by using a prior distribution for the stochasticity, the proposed MAESN method can adapt to new tasks much faster compared to general RL, MAML and learning latent spaces without fast adaption. One question is that in the experiments, the test task is from the training tasks with only dense reward being placed with sparse reward. During the training phrase, the model in MAESN could have already ‘memorized’ the tasks according to w_i so it is not surprising that MASEN outperforms MAML. I would like to see results if we use an entirely new task that is never seen by MAESN instead of changing the reward. Some minor problems: The references of equations in algorithm 1 are problematic where (3), (4), (5) should be (2), (3), (4). In the appendix, there is an empty ‘[]’.

Reviewer 3



This paper introduces a metalearning approach that discovers RL exploration strategies that work well across a particular distribution of MDPs. Furthermore, due to the metalearning approach used, the exploration behaviors can be adapted to the problem at hand quickly as the agent gains experience in the MDP. It is shown that this approach significantly outperforms other metalearning strategies such as MAML (which metalearns initial policy parameters), as well as random exploration. This appears to be due to the fact that the proposed method conditions on a per-episode latent variable that encourages temporally coherent exploration throughout the episode, rather than per-step randomness. Strengths: + The core idea of per-episode temporally-consistent exploration is excellent and much needed in the RL community. Per-step random search will never scale well to complex problems. This is an idea that will be useful to the community well beyond this particular instantiation of it. + Most exploration techniques are just glorified random or uniform exploration (e.g. intrinsic motivation, diversity metrics, etc). Instead, this approach uses the learned latent space to discover a space of useful exploratory behaviors. + Metalearning seems to be the right tool for this. The technical development of the MAESN algorithm is both sound and insightful, as is the analysis of why approaches that are limited to time-invariant noise don’t work. + The paper is written extremely clearly, being both technically thorough, and providing good high-level overviews of tricky concepts before diving it. + The evaluation is quite thorough as well. It is shown that the MAESN works significantly better than existing state-of-the-art approaches on a number of robot domains and that MAESN leads to sensible, temporally consistent exploration. There is also a solid intuitive explanation given of the results + analysis. + I liked that early “learning to learn” papers were cited in addition to more recent metalearning papers. Too often metalearning is talked about as though it is a brand new idea, so it is good to see some older works getting credit. Weaknesses: - No significant weaknesses. There could have been a larger diversity in domains (e.g. non-robotics domains such as scheduling, etc.). Other thoughts: - There may be something I’m missing, but I’m wondering why the KL divergence term was used instead of just clamping the pre-update w_i to the prior, since they converge to that anyway. Does it just make it harder to learn if there isn’t some flexibility there during learning? - There are a few other related papers that would be good to cite. This work is related in spirit to some prior work on searching for reward functions that speed up learning on a distribution of MDPs by essentially encouraging certain types of exploration. Some examples: @article{niekum2010genetic, title={Genetic programming for reward function search}, author={Niekum, Scott and Barto, Andrew G and Spector, Lee}, journal={IEEE Transactions on Autonomous Mental Development}, volume={2}, number={2}, pages={83--90}, year={2010}, publisher={IEEE} } @inproceedings{sorg2010internal, title={Internal rewards mitigate agent boundedness}, author={Sorg, Jonathan and Singh, Satinder P and Lewis, Richard L}, booktitle={Proceedings of the 27th international conference on machine learning (ICML-10)}, pages={1007--1014}, year={2010} }